# Smoking cessation and related factors in middle-aged and older Chinese adults: Evidence from a longitudinal study

Dechao Qiu[1,2], Ting Chen[1,2], Taiyi Liu[1,2], Fujian Song[3]*

**1** School of Public Health, Medical College, Wuhan University of Science and Technology, Wuhan, Hubei Province, China, **2** Hubei Province Key Laboratory of Occupational Hazard Identification and Control, Wuhan University of Science and Technology, Wuhan, Hubei Province, China, **3** Norwich Medical School, University of East Anglia, Norwich, Norfolk, United Kingdom

* Fujian.Song@uea.ac.uk

**Data Availability Statement:** The data that support the findings of this study are available in the China Health and Retirement Longitudinal Study at http://charls.pku.edu.cn.

## Abstract

### Objectives

There are more than 300 million smokers in China. This study aimed to evaluate the rate of smoking cessation, smoking relapse and related factors in middle-aged and older smokers in China.

### Methods

We performed a secondary analysis of data from China Health and Retirement Longitudinal Study (CHARLS) that recruited a nationally representative sample of adults aged 45 and older. Participants were 3708 smokers in 2011 who completed two waves of follow-up interviews in 2013 and 2015. Self-reported quit and relapse rates at follow-ups were estimated. Multiple logistic regressions were conducted to identify factors associated with smoking cessation and relapse.

### Results

The overall quit rate was 8.5% (95% CI 7.7% - 9.5%) at the 2-year follow-up in 2013, and 16.6% (95% CI 15.5% - 17.9%) at the 4-year follow up. Smoking cessation in 2013 was associated with not living in the northeast region ($p$ = 0.003), fewer cigarettes smoked daily ($p$ <0.001), and longer time to the first cigarette in the morning ($p$<0.001). Smoking cessation in 2015 was associated with older age ($p$ = 0.049), smoking initiation at age $\geq$20 years ($p$<0.001), longer time to the first cigarette in the morning ($p$<0.001), and self-perceived poor health ($p$<0.001). Of the 317 participants who stopped smoking in 2013, 13.3% (95% CI 9.9% - 17.5%) relapsed by 2015. Smoking relapse was associated with younger age ($p$ = 0.025), shorter time to the first cigarette in the morning ($p$ = 0.003), and self-perception of not poor health ($p$ = 0.018).

### Conclusion

The overall quit rate was 8.5% at the 2-year follow up, and 16.6% at the 4-year follow up in the middle-aged and older smokers, but 13% of quitters returned to smoking in two years.

**Funding:** The author(s) received no specific funding for this work.

**Competing interests:** The authors have declared that no competing interests exist.

Successful smoking cessation was associated with older age, lower nicotine dependence, and self-perceived poor health.

## Introduction

There are more than 300 million smokers, and the death toll from smoking-related diseases is estimated to exceed one million annually in China [1–3]. Tobacco control and smoking cessation are important actions to achieve the "Healthy China" official target [4].

Smoking cessation reduces tobacco-related harms regarding a variety of related diseases, including cancer, coronary heart disease, and chronic obstructive pulmonary disease [5, 6]. To promote smoking cessation, it is important to have an appropriate understanding of smoking cessation and its related factors. Previous studies reported that possible predictors of, or factors associated with, smoking cessation included age, socioeconomic status, health conditions, and the severity of nicotine dependence [7]. Evidence on smoking cessation and related factors is rare in China, particularly from longitudinal studies. A one-year follow-up study of participants of 'Quit and Win' in 2002 in China found that smoking cessation was associated with motivation to quit, age, and marital status [8]. China Seven Cities Study revealed that smoking abstinence was associated with lower nicotine dependence, perception of less stress and reduced hostility [9]. Findings from International Tobacco Control (ITC) China Survey identified older age, quitting intentions, nicotine dependence, and geographical location as important predictors of quit attempts or successful quitting [10].

However, previous studies of smoking cessation and related factors in China included only smokers from large cities. The present study used data from a large-scale longitudinal study of nationally representative samples to estimate the quit rates of current smokers, and to provide additional evidence on factors associated with successful smoking cessation in the middle-aged and older smokers.

## Methods

### Study design and sample

Data for this study came from China Health and Retirement Longitudinal Study (CHARLS) [11]. CHARLS is a longitudinal study that assessed the social, economic, and health status of a nationally representative sample of Chinese adults aged ≥45 years. The study covers 28 provinces, 150 county-level units and 450 village-level units and uses probability proportional to size (PPS) sampling. Regular follow-up with strict quality control maintained the representativeness of the middle-aged and elderly population, and formed a large-scale cohort data on aging problems in China [11].

The Biomedical Ethics Review Committee of Peking University approved the CHARLS study in January 2011. All participants were informed and provided informed consents before interviews. For this study, we applied to the CHARLS team and obtained anonymous data of participants.

The 2011 CHARLS survey provided baseline data. The follow-up rate of participants recruited in 2011 was 88% in 2013 and 87% in 2015. The present study included 3708 participants who were current smokers in 2011, and were followed up in both 2013 and 2015. We checked the data integrity, and excluded 109 participants with missing or abnormal data. Our study was a secondary analysis of data from the CHARLS.

## Measurements

Ever smokers in the CHARLS study were defined as participants who smoked more than 100 cigarettes in their lifetime, according to answers to the following question: "have you ever chewed tobacco, smoked a pipe, smoked self-rolled cigarettes, or smoked cigarettes/cigars?". Ever smokers were classified as current or former smokers (quitters) through the question "do you still have the habit or have you totally quit?". Current smokers in 2011 were re-interviewed in 2013 and classified into two categories: continued smokers or quitters in 2013. Current smokers in 2013 were classified as remained smokers and quitters in the 2015 survey, and quitters in 2013 were classified as quitters or relapsers in 2015 (Fig 1).

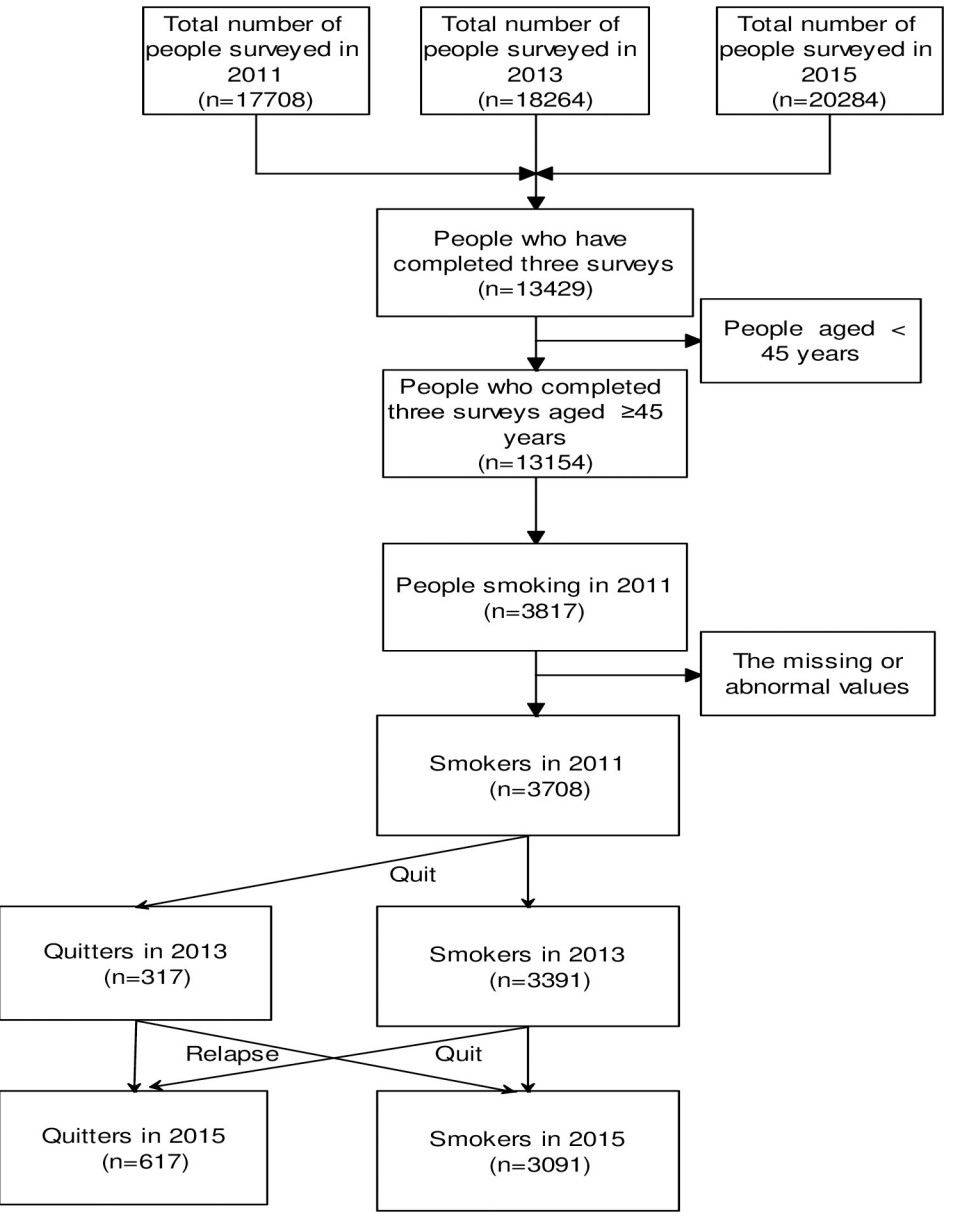

**Fig 1. Flow diagram for the process of participant selection and follow-up.**

The CHARLS study measured the level of nicotine dependence based on cigarettes per day, age at starting smoking, and time to the first cigarette after waking up. The CHARLS data also included the following information on participants: sociodemographic variables including age, sex, rural/urban location, educational level, marital status, region and self-perceived health condition. The perception of health by participants was measured by asking the following question: "Would you say your health is very good, good, fair, poor or very poor?"

## Statistical analysis

The quit rate was estimated by dividing the number of quitters at the end of the follow-ups in 2013 and 2015 by the number of current smokers before the follow-up in 2011 and 2013 (Fig 1). Therefore, quit rate in this study was self-reported point quit prevalence. To facilitate comparisons with different studies, we estimated the annualized quit rate ($q$) based on an assumption of constant quit rate during the follow-up period: $q = 1 - exp\{ln(1-nq/ns)/yr\}$, in which $ns$ refers to smokers before the follow-ups, $nq$ refers to the number of quitters at the end of follow-ups, and $yr$ refers to the number of follow-up years. The annualized quit rate had been used in previous studies with different years of follow-ups [12].

Between smokers and quitters, the differences in characteristics in the frequency of categorical variables were analyzed using the chi-squared test. Logistic regressions modeled the association between smoking cessation (dependent variable) and participant characteristics (independent variables). The selection of independent variables was restricted by data availability, and according to our understanding of factors that may be possibly associated with the dependent variable. We used the following independent variables: age, sex, rural/urban location, marital status, geographical region, educational level, self-perceived health status, cigarettes per day, age at starting smoking, and time to the first cigarette after waking up. Data were analyzed using Stata/MP 16 (Statacorp, TX). Two sided $p \leq 0.05$ was considered as statistically significant.

## Results

The 3,708 participants included 3323 males with an average age of 58.7 years, and 385 females with an average age of 61.5 years (Table 1). A total of 317 smokers reported that they stopped smoking in 2013, and 617 smokers stopped smoking in 2015 (Fig 1). The overall quit rate was 8.5% (95% CI: 7.7% - 9.5%) at the 2-year follow-up in 2013, and 16.6% (95% CI: 15.5% - 17.9%) at the 4-year follow-up, which corresponded to an annualized quit rate of 4.4%. There were 275 participants who quit smoking at both follow-ups in 2013 and 2015, with a quit rate of 7.4% (95% CI 6.6% - 8.3%). Of the 317 quitters in 2013, 42 returned to smoking again in 2015, with a relapse rate of 13.3% (95% CI: 9.9% - 17.5%) within two years.

Table 1 shows the quit rates by participant characteristics. Quitting in 2013 was significantly associated with longer time to the first cigarette after waking up, fewer cigarettes per day, and starting smoking at age ≥20 years. Smoking cessation in 2015 was significantly associated with older age, not living in the northeast region, longer time to the first cigarette after waking up, fewer cigarettes per day, and smoking initiation at age ≥20 years. Smoking cessation was significantly associated with deterioration of self-perceived health during 2011–2015. There were no significant associations between smoking cessation and other factors, including sex, rural or urban location, educational background, or marital status (Table 1).

The results of multivariable logistic regressions are shown in Table 2. Smoking cessation in 2013 was significantly associated with not living in the northeast region, fewer cigarettes smoked daily, and longer time to the first cigarette after waking up. For current smokers in 2013, smoking cessation in 2015 was significantly associated with older age, later age (≥20

**Table 1. Participant characteristics and smoking cessation rates.**

| Variables | N | 2013 | | 2015 | |
|---|---|---|---|---|---|
| | | Quit rate (%) (95% CI) | *p* | Quit rate (%) (95% CI) | *p* |
| Total | 3078 | 8.5 (7.7–9.5) | - | 10.1 (9.1–11.2) | - |
| Age(years) | | | | | |
| 45–54 | 1229 | 7.65(6.22–9.28) | 0.068 | 13.51(11.64–15.55) | <0.001 |
| 55–64 | 1543 | 8.68(7.32–10.20) | | 16.98(15.14–18.95) | |
| 65–74 | 745 | 8.86(6.92–11.13) | | 20.40(17.56–23.48) | |
| ≥75 | 191 | 12.04(7.79–17.52) | | 19.37(14.02–25.70) | |
| Sex | | | | | |
| Male | 3323 | 8.70(7.76–9.71) | 0.344 | 16.79(15.54–18.11) | 0.464 |
| Female | 385 | 7.27(4.89–10.34) | | 15.32(11.88–19.32) | |
| Rural/urban | | | | | |
| Rural | 2491 | 8.35(7.29–9.51) | 0.535 | 16.5(15.06–18.02) | 0.743 |
| Urban | 1217 | 8.96(7.41–10.70) | | 16.93(14.86–19.15) | |
| Educational status | | | | | |
| Illiteracy | 579 | 8.29(6.18–10.84) | 0.268 | 17.62(14.60–20.97) | 0.606 |
| Primary & junior high school | 2663 | 8.26(7.24–9.37) | | 16.45(15.06–17.91) | |
| High school & above | 466 | 10.52(7.88–13.66) | | 17.26(13.87–21.10) | |
| Marital status | | | | | |
| Married | 3342 | 8.65(7.72–9.65) | 0.517 | 16.91(15.65–18.22) | 0.188 |
| Other | 366 | 7.65(5.14–10.87) | | 14.21(10.80–18.21) | |
| Region | | | | | |
| East | 1167 | 9.00(7.42–10.79) | 0.376 | 17.91(15.75–20.23) | 0.046 |
| Central | 1035 | 9.03(7.35–10.95) | | 17.29(15.04–19.74) | |
| West | 1191 | 8.62(7.06–10.40) | | 16.20(14.16–18.42) | |
| Northeast | 315 | 6.03(3.67–9.26) | | 11.43(8.13–15.47) | |
| Health status in 2011 | | | | | |
| Poor | 1828 | 9.13(7.85–10.55) | 0.211 | - | - |
| Other | 1880 | 7.98(6.79–9.30) | | - | |
| Health status in 2013 | | | | | |
| Poor | 1802 | 9.10(7.81–10.52) | 0.243 | 18.31(16.55–20.18) | 0.008 |
| Other | 1906 | 8.03(6.85–9.34) | | 15.06(13.48–16.74) | |
| Health status in 2015 | | | | | |
| Poor | 1864 | 8.91(7.65–10.29) | 0.435 | 19.1(17.34–20.96) | <0.001 |
| Other | 1844 | 8.19(6.98–9.54) | | 14.15(12.59–15.83) | |
| Changes in self-perceived health status in two years | | | | | |
| Worse | 1184 | 8.36(6.85–10.09) | 0.78 | 18.51(16.36–20.82) | 0.033 |
| Other | 2524 | 8.64(7.57–9.80) | | 15.73(14.33–17.22) | |
| Time to the first cigarette after waking up | | | | | |
| ≤30 minutes | 2004 | 5.19(4.26–6.25) | <0.001 | 11.08(9.74–12.53) | <0.001 |
| >30 minutes | 1704 | 12.5(10.97–14.16) | | 23.18(21.20–25.26) | |
| Cigarettes per day | | | | | |
| <10 | 749 | 15.09(12.60–17.85) | <0.001 | 24.30(21.27–27.54) | <0.001 |
| 10–19 | 881 | 9.42(7.57–11.55) | | 18.62(16.10–21.35) | |
| ≥20 | 2078 | 5.82(4.85–6.91) | | 13.04(11.62–14.57) | |
| Age started smoking(years) | | | | | |
| <20 | 1299 | 7.01(5.68–8.54) | 0.014 | 13.09(11.30–15.04) | <0.001 |
| ≥20 | 2409 | 9.38(8.25–10.62) | | 18.56(17.03–20.18) | |

**Table 2. Results of logistic regressions analyses of factors associated with smoking cessation in 2013 and 2015.**

| Variables | Quit in 2013 | | Quit in 2015 | |
|---|---|---|---|---|
| | OR (95% CI) | *p* | OR (95% CI) | *p* |
| Age in 2011 | 1.01 (0.99–1.03) | 0.139 | 1.01 (1.00–1.03) | 0.049 |
| Sex (male vs. female) | 1.30 (0.84–2.00) | 0.237 | 1.43 (0.95–2.17) | 0.088 |
| Education (above vs. junior primary or below) | 1.19 (0.92–1.55) | 0.187 | 0.88 (0.68–1.14) | 0.331 |
| Married vs. other | 1.20 (0.79–1.84) | 0.390 | 1.37 (0.91–2.08) | 0.134 |
| Unban vs. rural | 1.08 (0.84–1.39) | 0.528 | 1.04 (0.82–1.33) | 0.730 |
| Northeast vs. others region | 0.40 (0.22–0.73) | 0.003 | 0.88 (0.57–1.35) | 0.544 |
| Age started smoking | 1.00 (0.99–1.02) | 0.565 | 1.02 (1.01–1.04) | <0.001 |
| Cigarettes per day | 0.97 (0.96–0.98) | <0.001 | 1.00 (0.99–1.01) | 0.747 |
| Smoking ≤30 minutes after waking up vs. >30 | 0.47 (0.37–0.61) | <0.001 | 0.54 (0.42–0.69) | <0.001 |
| Self-perceived poor health vs. other | 1.17 (0.92–1.48) | 0.203 | 1.61 (1.27–2.02) | <0.001 |

Notes to Table 2: Current smokers (n = 3708) in 2011 were included in the analysis for quitting in 2013, and continued smokers in 2013 (n = 3391) were included in the analysis for quitting in 2015.

years) at starting smoking, longer time to the first cigarette after waking up, and self-perceived poor health (Table 2).

Table 3 shows the results of the logistic regressions of factors associated with smoking relapse in the 317 quitters in 2013. Smoking relapse during 2013–2015 was associated with younger age (*p* = 0.024), shorter time to the first cigarette after waking up (*p* = 0.003), and self-perceived not poor health (*p* = 0.018). Smoking relapse was not significantly associated with sex, education, marital status, urban/rural location or region.

## Discussion

### Findings and comparison with other studies

The quit rate of current smokers in 2011 was 16.6% at the 4-year follow up in 2015, which corresponded to an annualized quit rate of 4.4%. This estimate of annualized quit rate was surprisingly similar to findings from previous large-scale longitudinal studies in China and in other countries, given the considerable differences in smokers' characteristics, socioeconomic and health care circumstances, and definition of smoking cessation outcomes. For example, a study in six cities in China found an annualized quit rate of 4.1% [10]. A study in the United States reported that 30.2% of the 5127 current smokers in 1993 stopped smoking by 2001, with

**Table 3. Results of multiple variable logistic regressions of factors associated with smoking relapse in 2015.**

| Variables | OR (95% CI) | *p* |
|---|---|---|
| Age in 2011 | 0.95 (0.91–0.99) | 0.024 |
| Sex (male vs. female) | 4.14 (0.51–33.39) | 0.183 |
| Education (above junior primary) | 0.76 (0.35–1.64) | 0.480 |
| Married vs. other | 0.35 (0.11–1.13) | 0.080 |
| Urban vs. rural | 0.95 (0.45–2.02) | 0.893 |
| Northeast vs. others region | 1.36 (0.25–7.26) | 0.722 |
| Age started smoking | 0.97 (0.92–1.02) | 0.204 |
| Cigarettes per day | 0.98 (0.95–1.02) | 0.388 |
| Smoking ≤30 minutes after waking up vs. >30 | 3.38 (1.53–7.47) | 0.003 |
| Self-perceived poor health vs. other | 0.41 (0.20–0.86) | 0.018 |

an annualized quit rate of 4.4% [12]. A 1-year follow-up study in 1996 in the UK reported a quit rate of 5% in smokers [13]. In a longitudinal study of 4636 smokers from seven centers in Northern Europe, the crude quit rate was 4.5 per 100 person-years [14]. In a Japanese study, the quit rate in 1358 smokers after 1-year follow-up was 4.6% (≥3-months' abstinence) or 3.1% (≥6 months' abstinence) [15].

The current study found that smoking cessation was associated with age, region, perception of health status, and nicotine dependence, which are similar to previous studies [10, 16]. Older smokers are more likely to quit smoking, which may be partially due to increased concerns about health [10], and the inverse U-shaped relationship of nicotine dependence with age [15]. Smokers in northeast China had a lower smoking cessation rate, which is similar to the findings of the ITC study and likely due to differences in tobacco culture across regions in China [10].

The smoking cessation rate was relatively high in smokers with self-perceived poor health. According to the previous studies, concern about personal health was a common reason for smokers to quit smoking [17]. When smokers realize that smoking is affecting their health, they may be more motivated to quit smoking [18]. Therefore, doctors and family members should take the opportunity to promote smoking cessation in patients with smoking related diseases.

Our results confirm findings from previous studies that smoking cessation was associated with tobacco dependence, including the number of cigarettes smoked daily, age of smoking initiation and time to the first cigarette in the morning [7]. Compared with participants who were less addicted, smokers with more severe nicotine dependence were less likely to quit [19, 20]. Adult smokers with more health problems may be more motivated to quit smoking than smokers with fewer health problems [21].

There are various interventions to help smokers quit smoking, as specified by the World Health Organization Framework Convention on Tobacco Control (FCTC) and MPOWER (Monitor, Protect, Offer, Warn, Enforce, Raise) [13]. Strong evidence revealed that policies of tobacco control intervention reduced the smoking rate of general population [22]. However, the implementation of the FCTC in China must be further strengthened, including political leadership, government oversight of the tobacco industry, and advocacy and support [23]. Smoking cessation support in China is rarely available, and only 5.6% smokers in China used smoking cessation medications [24]. Therefore, smoking cessation in China was predominantly unassisted compared to quitters in other countries [25]. Therefore, cessation treatments were unlikely to be an important factor that directly affected the quit rate or motivation to quit in the study participants.

We also found that smoking relapse was lower in older adults, smokers with less severe tobacco dependence, and who had perceived poor health. In addition to the self-perception of poor health, previous studies found that smoking relapse was associated with perceived stress, anxiety or depression [9, 26]. Further research is required to improve our understanding of smoking relapse and related factors.

## Strengths and limitations

The present study was based on data from a large, nationally representative longitudinal study that included urban and rural areas. Another strength of the study is the longitudinal data from two-waves of follow-ups, which provided rare opportunity for us to estimate quit rates and related factors in current smokers in China. However, this study has some limitations. First, the study relied on self-reported data and excluded participants with missing follow-up data, which may result in recall bias and errors. Second, we measured point prevalence of

smoking status at the two follow-ups in 2013 and 2015, and did not assess possible changes in smoking status between the assessments. Third, we conducted multiple statistical comparisons, and did not correct for possibly inflated type I errors. Furthermore, we did not consider the impacts of smoking cessation interventions (e.g., tobacco control policies, smoking cessation therapy) on smoking and quitting. These limitations should be appropriately addressed in further studies.

## Conclusion

The overall quit rate was 8.5% at the 2-year follow up, and 16.6% at the 4-year follow up in the middle-aged and older smokers, although 13% of quitters returned to smoking in two years. Successful smoking cessation was associated with older age, lower nicotine dependence, and self-perceived poor health.

## Acknowledgments

The authors thank the staff and participants of the China Health and Retirement Longitudinal Study (CHARLS) team for providing the data.

## Author Contributions

**Conceptualization:** Dechao Qiu, Ting Chen, Fujian Song.

**Investigation:** Dechao Qiu, Taiyi Liu.

**Methodology:** Dechao Qiu, Ting Chen, Taiyi Liu, Fujian Song.

**Writing – original draft:** Dechao Qiu.

**Writing – review & editing:** Dechao Qiu, Ting Chen, Taiyi Liu, Fujian Song.

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
