## [Decision Letter · Decision Letter 0]

2 Sep 2020

PONE-D-20-20662

Smoking cessation and related factors in middle-aged and older Chinese adults: evidence from a longitudinal study

PLOS ONE

Dear Dr. Song,

Thank you for submitting your manuscript to PLOS ONE. After careful consideration, we feel that it has merit but does not fully meet PLOS ONE’s publication criteria as it currently stands. Therefore, we invite you to submit a revised version of the manuscript that addresses the points raised during the review process.

The manuscript is informative and has the potential to make a contribution to the scientific literature.  However, major and significant revisions are required for the manuscript to meet the standards for publication.  Please attend to the English language deficits of the paper by having it thoroughly edited by a native English language user.  Please also attend to the all of the other comments made by the 3 reviewers including the issue of how the percentages of quitters are calculated and defined; the direction of the effect in statements of the results; and the attrition in the sample.  

We look forward to receiving your revised manuscript.

Kind regards,

Faith B. Dickerson

Academic Editor

PLOS ONE

Journal Requirements:

2. Your ethics statement must appear in the Methods section of your manuscript. If your ethics statement is written in any section besides the Methods, please move it to the Methods section and delete it from any other section. Please also ensure that your ethics statement is included in your manuscript, as the ethics section of your online submission will not be published alongside your manuscript.

Reviewers' comments:

Reviewer's Responses to Questions

**Comments to the Author**

1. Is the manuscript technically sound, and do the data support the conclusions?

Reviewer #1: Yes

Reviewer #2: Partly

Reviewer #3: Yes

2. Has the statistical analysis been performed appropriately and rigorously? 

Reviewer #1: Yes

Reviewer #2: Yes

Reviewer #3: Yes

3. Have the authors made all data underlying the findings in their manuscript fully available?

Reviewer #1: Yes

Reviewer #2: Yes

Reviewer #3: Yes

4. Is the manuscript presented in an intelligible fashion and written in standard English?

Reviewer #1: No

Reviewer #2: No

Reviewer #3: No

5. Review Comments to the Author

Reviewer #1: Qiu et al report smoking cessation and relapse data from a large, longitudinal study of middle-aged and older smokers in China. The authors evaluated a wide array of demographic, clinical and other variables that might potentially influence cessation and relapse rates. Factors that predicted cessation included geographic location, number of cigarettes smoked per day and time to first cigarette after waking; relapse prediction included age and poor overall health. Although the paper is difficult to follow at times due to language limitations, the study design is straightforward and impressively longitudinal, has a high power sample size and conveys valuable information regarding characterization of the smoking population in China. These data, in turn, may aid public health officials design and manage smoking cessation strategies.

Abstract

Please indicate in the abstract introduction what percentage of total smokers are those that are middle-aged and older.

The abstract (and entire paper) requires a thorough editing to correct word choice, grammar and punctuation. For example, the current word choice in the abstract makes it difficult to ascertain what are exactly the independent predictors for relapse: “being late smoking after waking up every day”?

Methods

“Ever smokers” are defined as greater than 100. This value seems excessively high.

How were health status and self-perceived poor health information defined/extracted?

Were data analyses corrected for multiple comparisons?

Results

Please include at the beginning of table 1 the overall quit rates for 2013 and 2015.

The tables and figure are all very informative.

The Discussion is comprehensive and appropriate, although again, the paper requires a writing service to correct the language problems.

Reviewer #2: This manuscript reports on a secondary analysis with the aim of identifying predictors of quitting and relapse in a sample of 3708 Chinese smokers who participated in a parent study (a general health epidemiology survey) in which they provided data on smoking status at baseline (2011), 2-year(2013), and 4-year (2015) follow ups.

This study makes a contribution by focusing on middle-aged smokers (> 45 yrs), which is useful since the over 40 yr old group represents increased health risk of continued smoking. Another novel aspect of the study is the representation of participants from rural China, which adds to the work that has been done in this area. Quit rates were approx. 10% at the 2 year follow up and 17% at the 4-year follow up with 86% of quitters at the 2 year follow up also reporting abstinence at the 4-year follow up. Predictors of quitting in multivariate analysis included older age, higher levels of nicotine dependence, and poorer self-rated health. This replicates other work done internationally as well as work done in China.

Suggestions:

As reported, I found the quit rates somewhat confusing. Since the authors say they will calculate quit rates as follows: “The quit rate was estimated by dividing the number of quitters at the follow-up survey by the number of smokers at baseline,” I think it would be useful to report the quit rate at 2013 as well as the quit rate at 2015 in terms of the entire sample, (617 self-reported being quit at the 2015 follow up, which would be a quit rate of 16.6% ) rather than just reporting the % quit at 2015 of those who were smoking at 2013.

An overall quit rate of 16.6% is reported in the manuscript from 2011-2015, is this meant to be the point prevalence self-reported quit rate at the 2015/4-year follow up? If yes would suggest changing the terminology to reflect this.

Between the 2 and 4 year follow-ups, a continuous quit rate of 7.4% (275/3708) was reported, however, was continuous quit rate actually assessed or could this be capturing people who happen to report being quit at the two assessments, but have not been continuously quit? If the latter is true, needs to be mentioned as a limitation that could be overestimating continuous quit rates. Also, although this does represent the “continuous” quit rate in the total sample, would suggest the authors emphasize the high % of quitters who maintained abstinence (86.7%) as well as noting, as they do, the relapse rate of 13.1% .

I do not understand what is meant here: “To facilitate the comparison with findings from different studies, the annual quit rate was estimated based on the quit rates for different follow-up years and an assumption of constant rates of still smoking during the follow-up.”

Two limitations that should be emphasized are the reliance on self-report data and on a completer analysis. Reliance on self-report data is mentioned in the discussion, but deserves more attention, particularly whether there are reasons people might lie about their smoking status (in the US, for example, people lie about this to health insurers because they do not want their rates to be higher). Statistical tests should be done to evaluate differences between those who completed both follow-ups to those who did not.

An alternative to age started smoking and number of cigs/day as separate predictor variables would be to create a composite variable “pack years.”

Did any unique predictors emerge to predict “continuous quit?” (Or are these all just the opposite of the predictors of relapse?)

Minor comments:

Throughout manuscript, when citing predictors of quitting, specify direction (e.g., age) of the effect

“The starting smoking age” should be rephrased “Age started smoking”

A few grammatical errors in this statement in the Abstract:

having less smoking number use, longer time to first cigarette upon waking. Independent predictors of staying quit in 2015 included being younger, being late smoking after waking up every day, having shorter smoking age

Reviewer #3: This is a study that examined smoking cessation, smoking relapse and related factors among middle-aged and older(≧45) smokers based on a secondary analysis of the dataset of the the China Health and Retirement Longitudinal Study (CHARLS). Here are some comments that may be helpful to improve the quality of this manuscript.

-The language needs to be further edited by native English speakers.

Abstract:

-The statistics of independent correlates of smoking quit, and staying quit should be provided in both the Abstract and the main text, such as p values, and ORs with 95%CI.

Methods:

-Statistics: Please specify which type of multiple logistic regression anayses were performed. Stepwise or other types? One- or two-sided tests? Clarify how to select the independent variables?

Results:

-Table 1: add (years) for Age.

Discussion:

-Strengths/Limitations: This is not a national survey.

-All Table/Figures should appear after the ref list, rather than in the text. This is general knowledge when drafting a manuscript.

6. PLOS authors have the option to publish the peer review history of their article (what does this mean?). If published, this will include your full peer review and any attached files.

Reviewer #1: No

Reviewer #2: **Yes: **Corinne Cather

Reviewer #3: No

---

## [Author Response · Author response to Decision Letter 0]

14 Sep 2020

Thanks for helpful comments from editors and peer reviewers. Please find our point-to-point response below. 

Editorial comments

Authors’ response: We have checked and revised the manuscript according to PLOS ONE's style requirements.

2. Your ethics statement must appear in the Methods section of your manuscript. If your ethics statement is written in any section besides the Methods, please move it to the Methods section and delete it from any other section. Please also ensure that your ethics statement is included in your manuscript, as the ethics section of your online submission will not be published alongside your manuscript.

Authors’ response: ethics statement added in the methods on page 5, paragraph 3. “The Biomedical Ethics Review Committee of Peking University approved the CHARLS study in January 2011. All participants were informed and provided informed consents before interviews. For this study, we applied to the CHARLS team and obtained data that maintained the participant anonymity.”

Review comments

Reviewer #1: Qiu et al report smoking cessation and relapse data from a large, longitudinal study of middle-aged and older smokers in China. The authors evaluated a wide array of demographic, clinical and other variables that might potentially influence cessation and relapse rates. Factors that predicted cessation included geographic location, number of cigarettes smoked per day and time to first cigarette after waking; relapse prediction included age and poor overall health. Although the paper is difficult to follow at times due to language limitations, the study design is straightforward and impressively longitudinal, has a high power sample size and conveys valuable information regarding characterization of the smoking population in China. These data, in turn, may aid public health officials design and manage smoking cessation strategies.

Authors’ response: Thank you very much for your positive comments. We have revised the manuscript and hope it is now satisfactory for being published in PLOS ONE.

Abstract

Please indicate in the abstract introduction what percentage of total smokers are those that are middle-aged and older.

Authors’ response: The CHARLS study recruited only participants aged 45 and older, so all smokers were middle-aged and older. We have clarified this in the manuscript’s Method section: “CHARLS is a longitudinal study that assessed the social, economic, and health status of a nationally representative sample of Chinese adults aged ≥45 years”(page 5, paragraph 2). 

The abstract (and entire paper) requires a thorough editing to correct word choice, grammar and punctuation. For example, the current word choice in the abstract makes it difficult to ascertain what are exactly the independent predictors for relapse: “being late smoking after waking up every day”?

Authors’ response: Thank for your comments. We have checked and clarified terms carefully, and used American Journal Experts (AJE) service to edit the English language. For example, we amended “being late smoking after waking up every day” to “longer time to first cigarette in the morning”. 

Methods

(1)“Ever smokers” are defined as greater than 100. This value seems excessively high. (2) How were health status and self-perceived poor health information defined/extracted? (3) Were data analyses corrected for multiple comparisons?

Authors’ response: Thanks for your comments. (1) We have clarified that “Ever smokers in the CHARLS study were defined as participants who smoked more than 100 cigarettes in their lifetime, according to answers to the following question: “have you ever chewed tobacco, smoked a pipe, smoked self-rolled cigarettes, or smoked cigarettes/cigars?” (page 6, para 2). (2) The perception of health by participants was measured by asking the following question: “Would you say your health is very good, good, fair, poor or very poor?” (page 7, para 1). (3) We did not attempt to correct for multiple comparisons, which has now been mentioned as a limitation in Discussion: “Third, we conducted multiple statistical comparisons, and did not correct for possibly inflated type I errors.” 

Results

Please include at the beginning of table 1 the overall quit rates for 2013 and 2015. The tables and figure are all very informative.

Authors’ response: Thank for this suggestion. We have added the total quit rates for 2013 and 2015 in Table 1. 

The Discussion is comprehensive and appropriate, although again, the paper requires a writing service to correct the language problems.

Authors’ response: We have revised the discussion by focusing on some unclear points, and received help on language problems from American Journal (AJE) service team. 

Reviewer #2: This manuscript reports on a secondary analysis with the aim of identifying predictors of quitting and relapse in a sample of 3708 Chinese smokers who participated in a parent study (a general health epidemiology survey) in which they provided data on smoking status at baseline (2011), 2-year(2013), and 4-year (2015) follow ups. This study makes a contribution by focusing on middle-aged smokers (> 45 yrs), which is useful since the over 40 yr old group represents increased health risk of continued smoking. Another novel aspect of the study is the representation of participants from rural China, which adds to the work that has been done in this area. Quit rates were approx. 10% at the 2 year follow up and 17% at the 4-year follow up with 86% of quitters at the 2 year follow up also reporting abstinence at the 4-year follow up. Predictors of quitting in multivariate analysis included older age, higher levels of nicotine dependence, and poorer self-rated health. This replicates other work done internationally as well as work done in China.

Authors’ response: Thanks for reviewer’s positive comments.

Suggestions:

As reported, I found the quit rates somewhat confusing. Since the authors say they will calculate quit rates as follows: “The quit rate was estimated by dividing the number of quitters at the follow-up survey by the number of smokers at baseline,” I think it would be useful to report the quit rate at 2013 as well as the quit rate at 2015 in terms of the entire sample, (617 self-reported being quit at the 2015 follow up, which would be a quit rate of 16.6% ) rather than just reporting the % quit at 2015 of those who were smoking at 2013.

Authors’ response: Thanks for this suggestion. We have now revised the manuscript to improve the understandability and corrected language errors. In the method section, we clarified the method to calculate the quit rate: “The quit rate was estimated by dividing the number of quitters at the end of the follow-ups in 2013 and 2015 by the number of current smokers before the follow-up in 2011 and 2013 (see Fig 1).” In Result section, quit rates at the two follow-ups were reported: “The overall quit rate was 8.5% (95% CI 7.7%-9.5%) at the 2-year follow-up in 2013, and 16.6% (95% CI 15.5%-17.9%) at the 4-year follow up.”

An overall quit rate of 16.6% is reported in the manuscript from 2011-2015, is this meant to be the point prevalence self-reported quit rate at the 2015/4-year follow up? If yes would suggest changing the terminology to reflect this.

Authors’ response: Indeed, the quit rates were based on point prevalence self-reported quit. We have clarified this in Method section: “The quit rate was estimated by dividing the number of quitters at the end of the follow-ups in 2013 and 2015 by the number of current smokers before the follow-up in 2011 and 2013 (Fig 1) . Therefore, quit rates in this study were point self-reported quit rates.” 

Between the 2 and 4 year follow-ups, a continuous quit rate of 7.4% (275/3708) was reported, however, was continuous quit rate actually assessed or could this be capturing people who happen to report being quit at the two assessments, but have not been continuously quit? If the latter is true, needs to be mentioned as a limitation that could be overestimating continuous quit rates. Also, although this does represent the “continuous” quit rate in the total sample, would suggest the authors emphasize the high % of quitters who maintained abstinence (86.7%) as well as noting, as they do, the relapse rate of 13.1% .

Authors’ response: Thanks for this comment. We have clarified this as a limitation: “Second, we measured point smoking cessation status at the two follow-ups in 2013 and 2015, and did not assess possible changes in smoking status between the assessments”. (page16, para 3) 

I do not understand what is meant here: “To facilitate the comparison with findings from different studies, the annual quit rate was estimated based on the quit rates for different follow-up years and an assumption of constant rates of still smoking during the follow-up.”

Authors’ response: Reported quit rates in different studies were often based on different follow-up periods, and were not directly comparable. In order to compare results of different studies, an estimate of annual quit rate as a common outcome is helpful. We have now added more details on method and the equation used to convert 2-year or 4-year quit rates to an annual quit rate in the manuscript: “To facilitate comparisons with different studies, we estimate the annual quit rate (q) based on an assumption of constant quit rate during the follow-up: q=1-exp⁡{ln⁡(1-nq/ns)/yr}, in which ns refers smokers at the beginning of follow-ups, nq refers the number of quitters at the end of follow-ups, and yr refers the number of follow-up years.” (page 7, para 2). 

Two limitations that should be emphasized are the reliance on self-report data and on a completer analysis. Reliance on self-report data is mentioned in the discussion, but deserves more attention, particularly whether there are reasons people might lie about their smoking status (in the US, for example, people lie about this to health insurers because they do not want their rates to be higher). Statistical tests should be done to evaluate differences between those who completed both follow-ups to those who did not.

Authors’ response: We agree with these helpful comments. We have amended “limitation” section, with more explanations: “First, the study relied on self-reported data and excluded participants with missing data on follow-ups, which may result in recall bias and errors.” (page16, para 3). Because of resource and time restrictions, we are unable to conduct further analyses to compare participants included and those excluded for missing data reasons. We mentioned that “these limitations should be appropriately addressed in further studies.” (page 15, para 2) 

An alternative to age started smoking and number of cigs/day as separate predictor variables would be to create a composite variable “pack years.”

Authors’ response: Thanks for suggesting the use of a composite variable as an alternative index of tobacco dependence. Because the effects of age of smoking initiation and cigs/day tended to be in the same direction, the use of a composite variable may not materially change the results of this study. 

Did any unique predictors emerge to predict “continuous quit?” (Or are these all just the opposite of the predictors of relapse?)

Authors’ response: Thanks for this interesting question. In deeded, continued smoking and relapse tended to be associated with similar predictors (such as age, tobacco dependence, and self-perceived health), although the statistical significances may change due to different sample/even sizes. 

Minor comments:

Throughout manuscript, when citing predictors of quitting, specify direction (e.g., age) of the effect

Authors’ response: Thanks for this helpful advice. We have revised the manuscript to indicate the direction of the effects, for example: “Smoking cessation in 2013 was associated with not living in the northeast region (P=0.003), fewer cigarettes smoked daily (P<0.001), and longer time to first cigarette in the morning (P<0.001)”. (Abstract) 

“The starting smoking age” should be rephrased “Age started smoking”

Authors’ response: Thanks for this advice. We have now used “age started smoking” or “age at starring smoking in the manuscript. 

A few grammatical errors in this statement in the Abstract: having less smoking number use, longer time to first cigarette upon waking. Independent predictors of staying quit in 2015 included being younger, being late smoking after waking up every day, having shorter smoking age

Authors’ response: Thanks for pointing out these errors. We have revised the abstract (and the whole manuscript) to correct any identified grammatical errors. The revised abstract result section: “The overall quit rate was 8.5% (95% CI 7.7% - 9.5%) at the 2-year follow-up in 2013, and 16.6% (95% CI 15.5% - 17.9%) at the 4-year follow up. Smoking cessation in 2013 was associated with not living in the northeast region (P=0.003), fewer cigarettes smoked daily (P<0.001), and longer time to first cigarette in the morning (P<0.001). Smoking cessation in 2015 was associated with older age (P=0.049), smoking initiation at age ≥20 years (P<0.001), longer time to first cigarette in the morning (P<0.001), and self-perceived poor health (P<0.001). Of the 317 participants who stopped smoking in 2013, 13.3% (95% CI 9.9% - 17.5%) relapsed by 2015. Smoking relapse was associated with younger age (P=0.025), shorter time to first cigarette in the morning (P=0.003), and self-perception of good health (P=0.018).”

Reviewer #3: This is a study that examined smoking cessation, smoking relapse and related factors among middle-aged and older(≧45) smokers based on a secondary analysis of the dataset of the the China Health and Retirement Longitudinal Study (CHARLS). Here are some comments that may be helpful to improve the quality of this manuscript. -The language needs to be further edited by native English speakers.

Authors’ response: Thank Reviewer #3 for helpful comments. We have revised the manuscript carefully. In addition, the manuscript has been proofread by a writing service by American Journal Experts (AJE).

Abstract:

-The statistics of independent correlates of smoking quit, and staying quit should be provided in both the Abstract and the main text, such as p values, and ORs with 95%CI.

Authors’ response: Thanks for these suggestions. We have revised the abstract and manuscript to provide details on p values, OR and 95% Cis where possible. Due to word count limitation, we reported 95%Cis for quit and relapse rates and p values for significant predictors in the Abstract. As an example, here is the revised abstract result: “The overall quit rate was 8.5% (95% CI 7.7% - 9.5%) at the 2-year follow-up in 2013, and 16.6% (95% CI 15.5% - 17.9%) at the 4-year follow up. Smoking cessation in 2013 was associated with not living in the northeast region (P=0.003), fewer cigarettes smoked daily (P<0.001), and longer time to first cigarette in the morning (P<0.001). Smoking cessation in 2015 was associated with older age (P=0.049), smoking initiation at age ≥20 years (P<0.001), longer time to first cigarette in the morning (P<0.001), and self-perceived poor health (P<0.001). Of the 317 participants who stopped smoking in 2013, 13.3% (95% CI 9.9% - 17.5%) relapsed by 2015. Smoking relapse was associated with younger age (P=0.025), shorter time to first cigarette in the morning (P=0.003), and self-perception of good health (P=0.018).” 

Methods:

-Statistics: Please specify which type of multiple logistic regression analyses were performed. Stepwise or other types? One- or two-sided tests? Clarify how to select the independent variables?

Authors’ response: We did not use stepwise analyses to select independent variables in multiple logistic regression analyses, and have fully reported the analysis results (statistically significant or not). We have clarified variable selection in Method section: “The selection of independent variables was restricted by data availability, and according to our understanding of factors that may be possibly associated with the dependent variable.” 

Results:

-Table 1: add (years) for Age.

Authors’ response: Thanks and we have added “years” for age in Table 1. (page10)

Discussion:

-Strengths/Limitations: This is not a national survey.

Authors’ response: CHARLS covers 28 provinces, 150 county-level units and 450 village-level units in China. The sampling methods aimed to have a nationally representative sample of Chinese adults aged ≥45 years. 

-All Table/Figures should appear after the ref list, rather than in the text. This is general knowledge when drafting a manuscript.

Authors’ response: Thanks for your comments. According to PLOS ONE’s style requirements, we have now removed Fig 1 from the manuscript (submitted separately as an independent image file). However, tables remain “directly after the paragraph in which they are first cited”.

---

## [Editor Report · Decision Letter 1]

22 Sep 2020

PONE-D-20-20662R1

Smoking cessation and related factors in middle-aged and older Chinese adults: Evidence from a longitudinal study

PLOS ONE

Dear Dr. Song,

Thank you for submitting your manuscript to PLOS ONE. After careful consideration, we feel that it has merit but does not fully meet PLOS ONE’s publication criteria as it currently stands. Therefore, we invite you to submit a revised version of the manuscript that addresses the points raised during the review process.

Please make the following changes so that your manuscript will be acceptable for publication. Note – that the comments below are with reference to the tracked changes resubmitted version

Abstract and throughout the manuscript – P for probability should be “p” and not “P”Abstract page 2 – line 31 and line 33: correct grammar to state – “longer time to the first cigarette” - note addition of “the”Page 3 line 38: Conclusions: Because data were only collected every two years, it is not feasible to state what % of persons quit smoking annually.  Please remove this statement from the abstract and also from other parts of the manuscript.Page 5 line 71 – delete “the”Page 6 line 86- change to “ participants’ “Page 6 – line 89 – delete the word “respectively”Page 6 line 91 – Please clarify – Participants were those who were followed up in BOTH 2013 and 2015?  If so, please statePage 7 – line 107 – persons who were quitters at both time points were not necessarily “continuous” quitters as the data were point prevalences.   They were quitters at both time points.Page 7 line 115 – here and elsewhere use the term “sex” and not “gender”Page 8 – please delete text about how the annual quit rate was calculated.  If persons were queried every 2 years, it is not known what is the annual quit rate.Page 8 line 128 and 129 – if this text is to be retained note that the correct grammar is “…refers to the number of ….”Page 8 line 133 – should be “Logistic regressions” (note plural)Page 8 line 138 – last word on page and going to next page– note that “health status” is cited twicePage 9 – line 155 – please delete results about the annual quit ratePage 10 – line 158 – I don’t think you can say that the quit rate was “continuous” – just that quit status was reported at both time pointsPage 10 – line 162 – remove the word “ statistically” and just say “was significantly associated”Page 10 – line 164 – the same issue as abovePage 10 – line 168.  Is the variable “deterioration of self-perceived health” or just “self perceived health”Page 10 – line 170 – replace “gender” with “sex”Page 12 – line 175 – should be “logistic regressions” (plural)Page 14 – line 193 – is the variable “good health” or is it “not poor health”?Page 14 – line 196, should be “or” not “and”Page 14 – line 204 and continuing to next page – remove reference to annual quit rate and also the discussion of the annual quit ratePage 16 – line 233, insert a “the” – should read “time to the first cigarette in the morning”Page 17 – line 245 – delete “to” – should read “must be further strengthened”Page 17 – line 248 – typo- note spelling - should be “medications”Page 19 – line 286 – eliminate statement about annual quit rate

We look forward to receiving your revised manuscript.

Kind regards,

Faith B. Dickerson

Academic Editor

PLOS ONE

---

## [Author Response · Author response to Decision Letter 1]

26 Sep 2020

Response to comments (R2)

Dear Dr Dickerson

Thank you very much for your swiftly completing the review of our revised manuscript (R1), with very detailed comments. We have revised the manuscript accordingly, and hope it is now satisfactory for being published in PLOS ONE. Please see our point-by-point response to the comments below.

1. Abstract and throughout the manuscript – P for probability should be “p” and not “P”

Authors’ response: We have used “p” to replace “P” in abstract and throughout the manuscript.

2. Abstract page 2 – line 31 and line 33: correct grammar to state – “longer time to the first cigarette” - note addition of “the”

Authors’ response: Thanks for pointing out this error. We have added “the” where required.

3. Page 3 line 38: Conclusions: Because data were only collected every two years, it is not feasible to state what % of persons quit smoking annually. Please remove this statement from the abstract and also from other parts of the manuscript.

Authors’ response: We have removed annual quit rate from the abstract. Please see our detailed response to comment point #10 about “annual quit rate”. 

4. Page 5 line 71 – delete “the”

Authors’ response: Thanks for this advice. We have deleted “the” from this sentence. 

5. Page 6 line 86- change to “participants”

Authors’ response: Thanks, we have revised “participant” as “participants”. 

6. Page 6 – line 89 – delete the word “respectively”

Authors’ response: Thanks for this suggestion. We have deleted the word “respectively”.

7. Page 6 line 91 – Please clarify – Participants were those who were followed up in BOTH 2013 and 2015? If so, please state

Authors’ response: Thanks for your comments. We have added “both” before “2013 and 2015”.

8. Page 7 – line 107 – persons who were quitters at both time points were not necessarily “continuous” quitters as the data were point prevalences. They were quitters at both time points.

Authors’ response: Thanks for this suggestion. We have deleted “continuous” before “quitters”.

9. Page 7 line 115 – here and elsewhere use the term “sex” and not “gender”

Authors’ response: We have adopted the recommendation and changed “gender” to “sex” in the manuscript.

10. Page 8 – please delete text about how the annual quit rate was calculated. If persons were queried every 2 years, it is not known what is the annual quit rate.

Authors’ response: Thanks but we disagree with this suggestion. Reported quit rates in different studies were often based on different follow-up periods. In order to compare results of different studies, we used survival analysis method to estimate annual quit rate as a common outcome. Otherwise, it would be impossible to compare results of different epidemiological studies. We believe our analysis approach is methodological sound and practically helpful. Your reason for removing the annual quit rate was that participants were followed up every 2 years. Actually the annual rate could be estimated based on 2 or more follow-up years. For example, a previous study had estimated “the annualized quit rate” in a cohort of current and former smokers followed over 13 years (reference: Hyland, et al. Predictors of cessation in a cohort of current and former smokers followed over 13 years. Nicotine Tob Res 2004 Dec;6 Suppl 3:S363-9.). Therefore, we would like to retain contents related to annual quit rate in the manuscript. We have added Hyland et al’s study as a reference in Method section, and used term “annualized quit rate” to avoid possible misunderstanding of “annual quit rate” in the manuscript. We have used the annualized quite rate mainly in the discussion to compare with results of other studies, and have deleted it from the abstract and conclusion. 

11. Page 8 line 128 and 129 – if this text is to be retained note that the correct grammar is “…refers to the number of ….”

Authors’ response: Thanks for pointing out this error. We have added “to” after “refers”.

12. Page 8 line 133 – should be “Logistic regressions” (note plural)

Authors’ response: Thank you, we have amended “logistic regression” to “logistic regressions” throughout the manuscript.

13. Page 8 line 138 – last word on page and going to next page– note that “health status” is cited twice

Authors’ response: Thanks for this suggestion. We have deleted the repeated "health status"

14. Page 9 – line 155 – please delete results about the annual quit rate

Authors’ response: Thanks but please see our response to comment point #10.

15. Page 10 – line 158 – I don’t think you can say that the quit rate was “continuous” – just that quit status was reported at both time points

Authors’ response: Thanks for this suggestion. We amended “continuous quit rate” to “quit rate”.

16. Page 10 – line 162 – remove the word “ statistically” and just say “was significantly associated”

Authors’ response: Thanks for this suggestion. We amended to “was significantly associated”.

17. Page 10 – line 164 – the same issue as above

Authors’ response: Thanks, and we amended to “was significantly associated”.

18. Page 10 – line 168. Is the variable “deterioration of self-perceived health” or just “self- perceived health”

Authors’ response: It was the variable “deterioration of self-perceived health”. When smokers perceived the deterioration of their health，they are more likely to change their smoking behavior.

19. Page 10 – line 170 – replace “gender” with “sex”

Authors’ response: Thanks, and we have used “sex” to replace “gender” in the manuscript. 

20. Page 12 – line 175 – should be “logistic regressions” (plural)

Authors’ response: Thanks, and we have amended “logistic regression” to “logistic regressions” throughout the manuscript.

21. Page 14 – line 193 – is the variable “good health” or is it “not poor health”?

Authors’ response: Thanks for this suggestion. We have amended to "not poor health".

22. Page 14 – line 196, should be “or” not “and”

Authors’ response: Thanks for pointing out these errors. We have amended to “or”

23. Page 14 – line 204 and continuing to next page – remove reference to annual quit rate and also the discussion of the annual quit rate

Authors’ response: Thanks, but we disagree with this suggestion. Please see our response to comment point #10. 

24. Page 16 – line 233, insert a “the” – should read “time to the first cigarette in the morning”

Authors’ response: Thanks, and we have added “the” in the relevant place in the manuscript.

25. Page 17 – line 245 – delete “to” – should read “must be further strengthened”

Authors’ response: Thanks for this helpful advice. We have deleted “to” here.

26. Page 17 – line 248 – typo- note spelling - should be “medications”

Authors’ response: Thanks for this suggestion. We have amended the spelling.

27. Page 19 – line 286 – eliminate statement about annual quit rate

Authors’ response: Thanks, and we have deleted “annual quit rate” from the conclusion.

---

## [Editor Report · Decision Letter 2]

30 Sep 2020

PONE-D-20-20662R2

Smoking cessation and related factors in middle-aged and older Chinese adults: Evidence from a longitudinal study

PLOS ONE

Dear Dr. Song,

Thank you for submitting your manuscript to PLOS ONE and for the detailed changes that you have made. After careful consideration, we feel that it has merit but does not fully meet PLOS ONE’s publication criteria as it currently stands. Therefore, we invite you to submit a revised version of the manuscript that addresses the points raised during the review process.

Please note below a few minor revisions, most related to English-language usage, that are required before the manuscript is fully acceptable.  These notes are with reference to the CLEAN version of the resubmitted manuscript.

Page 5, line 81 – should be “participants”

Page 7 – line 110 – should be “point prevalence” [add “prevalence”].  Here and elsewhere - page 16 - line 227.

Page 8, line 135 – delete “the” near end of the line

Page 11, row 152 – add “the” to “not living in the northeast region”

Page 12, line 162 – add “the: to “relapse in the 317 quitters”

Page 15, line 206 – typo at end of line – “the”

Page 15 – line 216 – add “had” to “…and who had perceived poor health”

Page 16 – line 223 – isn’t it two waves of follow-up not three?  Please correct here and elsewhere

We look forward to receiving your revised manuscript.

Kind regards,

Faith B. Dickerson

Academic Editor

PLOS ONE

---

## [Author Response · Author response to Decision Letter 2]

1 Oct 2020

Please note below a few minor revisions, most related to English-language usage, that are required before the manuscript is fully acceptable. These notes are with reference to the CLEAN version of the resubmitted manuscript.

1 Page 5, line 81 – should be “participants”

Authors’ response: Thanks, we have revised to “anonymous data of participants” and amended “participant” to “participants”.

2 Page 7 – line 110 – should be “point prevalence” [add “prevalence”]. Here and elsewhere - page 16 - line 227.

Authors’ response: We agree with these helpful comments. We have made changes in Page 7 – line 110, and in page 16 - line 227.

3 Page 8, line 135 – delete “the” near end of the line

Authors’ response: Thanks for this advice. We have deleted “the” from this sentence. 

4 Page 11, row 152 – add “the” to “not living in the northeast region”

Authors’ response: Thanks for pointing out this error. We have added “the” where required.

5 Page 12, line 162 – add “the: to “relapse in the 317 quitters”

Authors’ response: Thanks, and we have added “the” in the manuscript.

6 Page 15, line 206 – typo at end of line – “the”

Authors’ response: Thanks for this helpful advice. We have amended to “the smoking rate of general population”. 

7 Page 15 – line 216 – add “had” to “…and who had perceived poor health”

Authors’ response: We have adopted the recommendation and changed to “…and who had perceived poor health” in the manuscript.

8 Page 16 – line 223 – isn’t it two waves of follow-up not three? Please correct here and elsewhere

Authors’ response: Thanks for this suggestion. We have amended to “two waves of follow-up”.

---

## [Editor Report · Decision Letter 3]

5 Oct 2020

Smoking cessation and related factors in middle-aged and older Chinese adults: Evidence from a longitudinal study

PONE-D-20-20662R3

Dear Dr. Song,

We’re pleased to inform you that your manuscript has been judged scientifically suitable for publication and will be formally accepted for publication once it meets all outstanding technical requirements.

Kind regards,

Faith B. Dickerson

Academic Editor

PLOS ONE
---

## [Editor Report · Acceptance letter]

7 Oct 2020

PONE-D-20-20662R3 

Smoking cessation and related factors in middle-aged and older Chinese adults: Evidence from a longitudinal study 

Dear Dr. Song:

I'm pleased to inform you that your manuscript has been deemed suitable for publication in PLOS ONE. Congratulations! Your manuscript is now with our production department. 

Kind regards, 

on behalf of

Dr. Faith B. Dickerson 

Academic Editor

PLOS ONE